# Fiber-Optic System for Intraoperative Study of Abdominal Organs during Minimally Invasive Surgical Interventions

Ksenia Kandurova [1,*], Viktor Dremin [1], Evgeny Zherebtsov [1,2], Elena Potapova [1], Alexander Alyanov [1,3], Andrian Mamoshin [1,3], Yury Ivanov [4], Alexey Borsukov [5] and Andrey Dunaev [1]

[1]  Orel State University, Research and Development Center of Biomedical Photonics, 302026 Orel, Russia; v.dremin@oreluniver.ru (V.D.); evgenii.zherebtsov@oulu.fi (E.Z.); e.potapova@oreluniver.ru (E.P.); a.alyanov@oreluniver.ru (A.A.); dr.mamoshin@mail.ru (A.M.); dunaev@bmecenter.ru (A.D.)

[2]  University of Oulu, 90014 Oulu, Finland; evgenii.zherebtsov@oulu.fi

[3]  Orel Regional Clinical Hospital, 302028 Orel, Russia

[4]  Federal Scientific and Clinical Center for Specialized Medical Service and Medical Technologies, 115682 Moscow, Russia; ivanovkb83@yandex.ru

[5]  Problem Research Laboratory "Diagnostic Researches and Mini-invasive Technologies", Smolensk State Medical University, 214019 Smolensk, Russia; bor55@yandex.ru

*  Correspondence: k.kandurova@oreluniver.ru; Tel.: +7-910-268-29-46

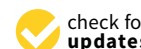

**Featured Application:** **The target audience of this application-oriented study are researchers interesting in the fields of the optically-guided surgery and optical biopsy. The paper features the experimentally obtained data with the endogenous fluorescence and tissue diffuse reflectance spectra from series of in vivo measurements in patients with pathologies of hepatoduodenal area. The results have applications and can be of particular interest in the development of new instruments for abdominal guided surgery.**

**Abstract:** The paper presents the results of experimental measurements of endogenous fluorescence and blood perfusion in patients with pathology of the organs of hepatopancreatoduodenal area in vivo. A custom setup combining channels for fluorescence spectroscopy (excitation wavelengths of 365 nm and 450 nm) and laser Doppler flowmetry (1064 nm) with fibre optical probe for nondestructive laparoscopic measurements has been developed and applied during minimally invasive operation procedure. Preliminary measurements with two aforementioned channels have been performed at specified excitation wavelengths. The possibility of obtaining fluorescence spectra and laser Doppler flowmetry signals in vivo during minimally invasive interventions was shown. Obtained data show perspectives of further research on technical and methodological development of optical diagnostic methods for minimally invasive surgery. The obtained results can be used to provide a deeper understanding of pathological processes influence on optical properties of abdominal organs tissues, which will ultimately help surgeons to determine the state of vitality in tissues and mucous membranes directly during the process of surgical intervention.

**Keywords:** optical biopsy; optical non-invasive diagnostics; fluorescence spectroscopy; laser Doppler flowmetry; minimally invasive surgery; hepatopancreatoduodenal area

## 1. Introduction

Despite the high technical and intellectual level of modern medicine, the problem of diagnosis and treatment of hepatopancreatoduodenal organs pathologies is still acute. The significance of this

issue is caused by high rate of morbidity, great risk of complications together with steep mortality rate [1,2].

In recent decades strong interest from research and clinical community is shown towards the introduction of minimally invasive technologies and techniques for diagnosis and treatment in clinical practice. This is caused by decreased surgical injury and the volume of surgical interventions, as well as the substantially reduced number of postoperative complications. This approach features significantly reduced time required for treatment and recovery, especially compared with open surgery [3].

During the minimally invasive operation, the choice of a rational medical tactics strictly depends on the observed syndrome, but usually suffer from lack of information during the procedure, as the surgical site is limited by the field of view of the optical channel. Currently, the integral parts of any minimally invasive intervention are the methods of ultrasound and X-ray examination. They are necessary for determining the allocation of pathological formations (such as neoplasms or abscesses), as well as more complex methods of computer tomography (CT) and magnet resonance imaging (MRI). However, these methods mostly show anatomical and morphological features at the organ level whereas in most cases it is important to assess the type and vitality of tissues at the cellular level. Therefore, obtaining of additional diagnostic information about the properties of the tissues intraoperatively is of particular interest for modern surgery. Speaking of CT and MRI these methods cannot be used immediately during the surgery. Another problem of X-ray and CT methods to be worried about is the harmful effects of the radiation.

A gold standard for analysis of state of biological tissues is histological examination, but it takes time to obtain the results. The development of minimally invasive technologies determines the need for development of diagnostic methods that allow for obtaining information in vivo with accuracy comparable to histological data. The implementation of new methods will increase the quality of diagnosis of biological tissues state and nature (healthy, inflamed, malignant, etc.) [4], and consequently improve the quality of treatment.

Optical diagnostic methods have been used for many years in various fields of medicine. Currently, the studies of application of spectrometry and imaging in surgery are quickly developing worldwide [5]. One of such methods is fluorescence spectroscopy (FS). The method is sensitive both to the emission of administered fluorescence contrast [6] as well as endogenous fluorescence of certain tissue biomarkers in vivo [7–9]. FS is aimed to studying the structure of cells, their properties and ongoing biochemical reactions. In certain cases, such parameters of fluorescence as intensity and lifetime depend on the metabolism conditions in the biological tissues under study. In particular, it is possible to evaluate cell metabolism by studying the ratio of nicotine adenine dinucleotide (NADH) and flavinoadenidine dinucleotide (FAD) coenzymes. These compounds are vital elements of tricarbonic acid cycle. Changes in the content of NADH and FAD influence the intensity of endogenous fluorescence excited by radiation of certain wavelengths. Thus, FS method creates an opportunity for diagnosis and monitoring of tissue metabolic activity in inflammatory processes [10,11]. Currently fluorescent diagnostics is successfully applied in several areas of medicine [12,13].

Normal level of blood perfusion in abdominal tissues is highly important for normal oxygenation and nutrition of them at the cellular level. Ischemia, or lack of perfusion, is a common way for tissue death or degeneration. Under minimally invasive surgical interventions, measurement of the microcirculation index in pathological tissues may be necessary to assess the relative contribution of ischemia to disease process. The method of laser Doppler flowmetry (LDF) is based on the evaluation of tissue perfusion by blood by probing with monochromatic laser radiation, followed by recording and analyzing changes in the spectrum of the reflected radiation. LDF provides continuous and real-time measurements of microvascular blood flow parameters and is highly sensitive to rapid changes in perfusion in capillary circulation. It has proved itself to be a method that can be included in multimodal diagnostic devices, and specifically in surgical operations on the organs of the digestive tract [14].

In this study, we have combined FS and LDF measurement techniques in one fibre optical probe to record the parameters of endogenous fluorescence and blood perfusion during minimally invasive interventions [15–17]. Thus, the overall aims of this work were to test the technical implementation of such measurements as well as register reference patterns of the fluorescence spectra and blood perfusion time series in organs of hepatopancreatoduodenal zone.

## 2. Materials and Methods

A fiber-optic system implementing both fluorescence spectroscopy and laser Doppler flowmetry methods was developed (Figure 1a) especially for performing experimental measurements from approximately one diagnostic volume of biotissue (1–3 mm$^3$). The main units of device were designed in cooperation with SPE "LAZMA" Ltd. (Moscow, Russia).

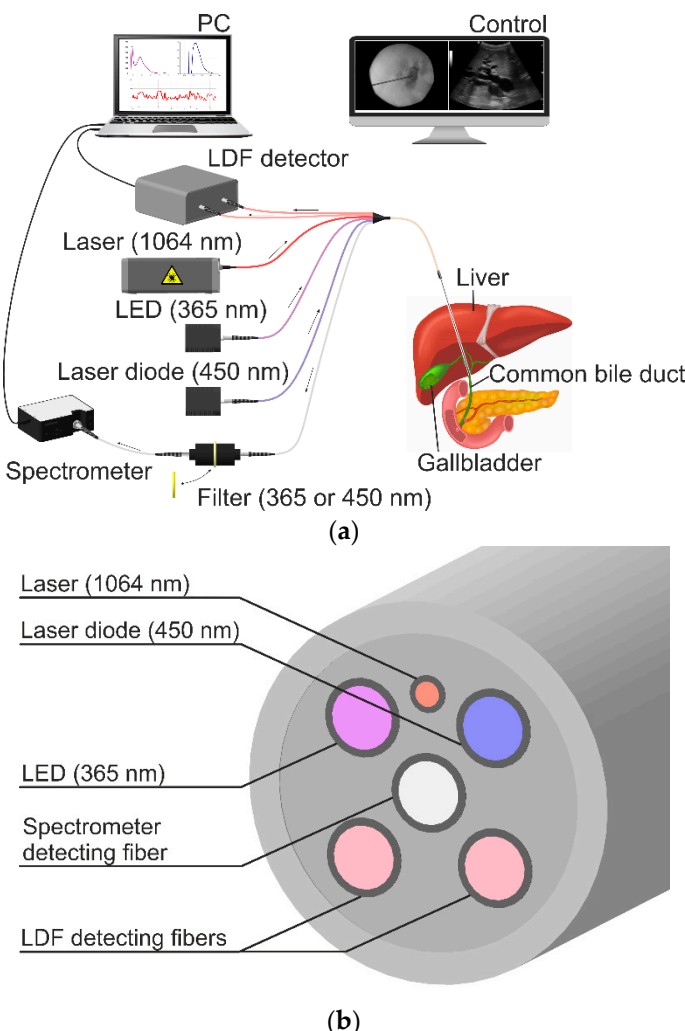

**Figure 1.** Schematic presentation of installation for experimental studies (**a**) and location of fibers in the laparoscopic probe (**b**).

A distinctive feature of the system is laparoscopic optical probe (30 cm in length and 3 mm in diameter), designed to access organs under investigation through standard instruments for minimally invasive manipulations. The probe contains 6 optical fibers (Figure 1b). A single-mode laser module with a radiation wavelength of 1064 nm was used in LDF channel, sources of 365 nm and 450 nm wavelengths were used for fluorescence excitation. The wavelengths were selected due to the fluorescent properties of such main endogenous fluorophores as NADH, FAD and stromal collagen. Namely, reduced NADH has maximum of emission at approx. 490 nm under excitation with UV

radiation (365 nm). Oxidized FAD fluoresces with peak of emission at approx. 520–540 nm under excitation radiation with wavelength of 450 nm [18].

The probing fiber of LDF channel has diameter of 6 µm, 2 receiving fibers have a diameter of 400 µm. The source-detector distance for LDF channel is 1.5 mm. The diameters of the probing and receiving fibers of FS channel are 400 µm. The numerical aperture of the fibers is 0.22. For safety reasons, as well as to minimize photobleaching effect, the radiation power for a wavelength of 365 nm at the fiber probe output does not exceed 1.5 mW. To assess the safety of the optical probe, effective tissue irradiance was calculated and its value was about 1.4 W/m$^2$. According to the guidelines of the International Commission on Nonionizing Radiation Protection (ICNIRP), the limiting UV exposure duration per day for this level of effective irradiance is at least 10 s. In the study, the actual time of the UV-irradiation did not exceed 2 s. [19]. Therefore, the using of proposed source wavelength was deemed safe. The output power for an excitation wavelength of 450 nm does not exceed 3.5 mW. The distance between the radiation source and receiver in FS channel is 1 mm.

A spectrometer of 350–820 nm detection range is used to record the fluorescence spectra. Data from the device is transferred to a personal computer for storing and further processing.

The studies were conducted at the department of interventional radiology of Orel Regional Clinical Hospital (Orel, Russia). This part of the work involved 42 patients aged 68 ± 12. Areas of interest included three types of organs with following pathologies: common bile duct (obstructive jaundice caused by pancreatic cancer or cholelithiasis, 29 patients); gallbladder (acute destructive cholecystitis, 5 patients); liver abscess (8 patients). All studies were approved by the local Committee for Human Biomedical Research Ethics (record of the meeting №10 of 16.11.2017) and were carried out in accordance with the principles outlined in the 2013 Declaration of Helsinki by the World Medical Association. The patients signed an informed consent indicating their voluntary willingness to participate in the study. The measurements were performed during diagnostic and therapeutic interventions under ultrasound and X-ray examination. The fibre optical probe was inserted into the lumen of the hollow organs (gallbladder, common bile duct) via minimally invasive approach. After that, the double excitation wavelengths autofluorescence has been recorded in several places from both affected and unaffected by the pathology areas.

## 3. Results and Discussion

The FS spectra obtained during the measurements were normalized by the level of backscattered excitation radiation. The results of averaging these spectra are shown in Figure 2.

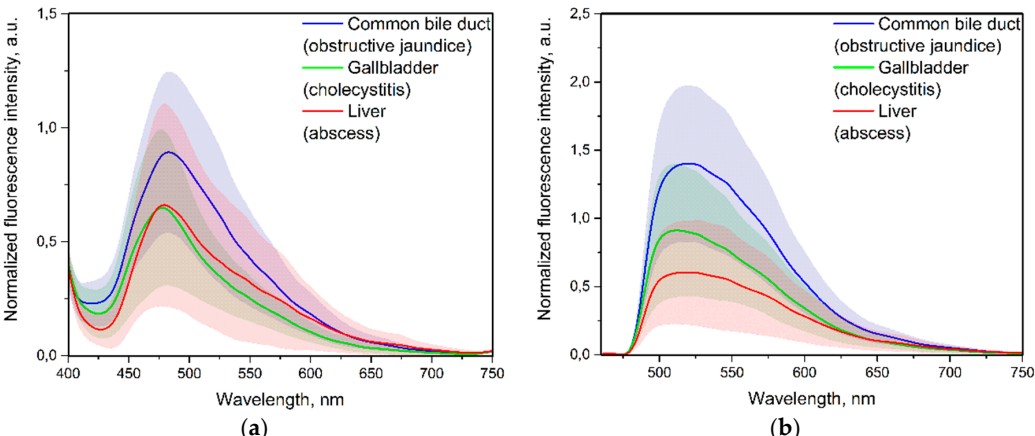

**Figure 2.** Averaged normalized fluorescence spectra under excitation wavelengths 365 nm (**a**) and 450 nm (**b**).

The main feature of the collected data is a high variability of fluorescence intensities. It is assumed that this is caused by both individual variability and different factors, such as, the state and nature

of changes in tissues, the phase development of pathological processes as well as ongoing treatment. In some patients, it was difficult to register fluorescence spectra with sufficient level of signal to noise ratio, which may be due to the presence of bile, blood and pus, which absorbed major part of fluorescence emission [20]. The components of blood, especially haemoglobin, are good absorbers of optical radiation and one of possible measures to reduce this effect is applying to the probe a moderate pressure. This technique allows one to decrease the blood content in the tissue as well as alleviate the attenuation effect of blood on the fluorescence spectrum [21]. However, in surgery in many cases the pressure applied to the end of optical probe can have negative effects on the soft tissues under study. The problem of accurate measurements requires an appropriate solution. The use of additional optical methods (e.g., diffuse reflectance spectroscopy) can be considered. Extending the number of implemented modalities allows for obtaining the data, which can be used to compensate the influence of blood on recorded fluorescence spectra [22]. If the blood content of the living tissue is to be kept unaffected, then to eliminate the probe pressure, an embedded in the probe noncontact distance measurements with subsequent automation of the recording procedure at a fixed optimal distance can be one of the possible solutions. Moreover, another measure to increase the level of signal to noise ratio is automation of mucus, pus and blood removal from the surface of optical probe during the procedure with minimal impact on the surface of biological tissue.

Obtained spectral data were compared and assessed for statistically significant differences using one-way analysis of variance (ANOVA). Statistically significant difference between averaged levels of peak fluorescence intensity in the types of areas under study was observed (Figure 3).

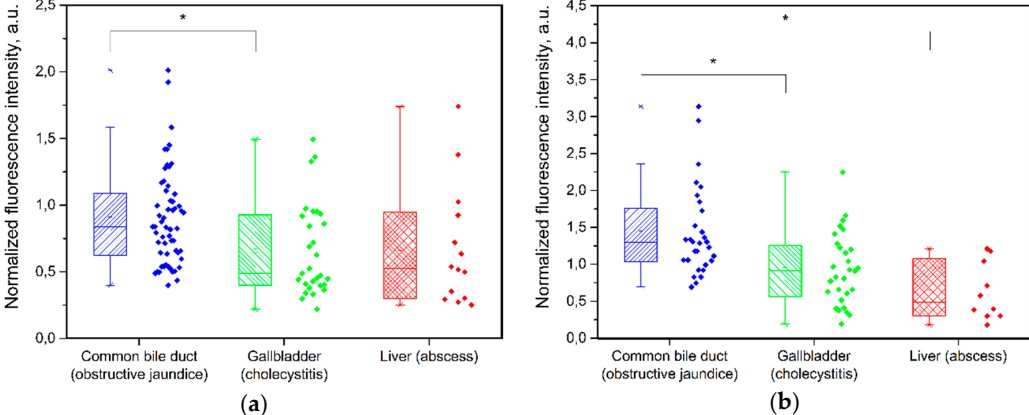

**Figure 3.** Box-Whisker diagrams of fluorescence intensity maxima under excitation wavelengths 365 nm (**a**) and 450 nm (**b**). * Confirmed statistically significant differences. The level of significance was estimated using a one-way analysis of variance ($p < 0.05$)

It was noted that the normalized values of fluorescence intensities are higher in common bile duct walls compared with the data obtained in the other areas. Normalized fluorescence intensities under 450 nm excitation wavelength was higher than the ones under 365 nm. The significant differences between fluorescence maxima in gallbladder and liver abscess with excitation at 450 nm were noticed as well (Table 1). This can be related to the pathological changes in the tissues including acute inflammatory process in abscess formation.

**Table 1.** Averaged fluorescence intensity maxima.

| Area | Mean | | Standard Deviation | |
|---|---|---|---|---|
| | 365 nm | 450 nm | 365 nm | 450 nm |
| Common bile duct | 0.90 | 1.43 | 0.36 | 0.61 |
| Gallbladder | 0.66 | 0.93 | 0.34 | 0.49 |
| Liver | 0.67 | 0.63 | 0.45 | 0.39 |

Examples of LDF signals recorded in the different areas of interest are shown in Figure 4a. Recordings of LDF signals were associated with several problems such as anatomical features preventing accurate positioning of the probe in interested area and holding it for a long time, as well as some level of inconvenience for patients causing movements artefacts. This affected the length of registered signals. The means of every signal (Table 2) were obtained to analyze for statistically significant differences by ANOVA. The results of analysis are shown in Figure 4b. Statistically significant differences were observed between common bile duct and remained areas of interest.

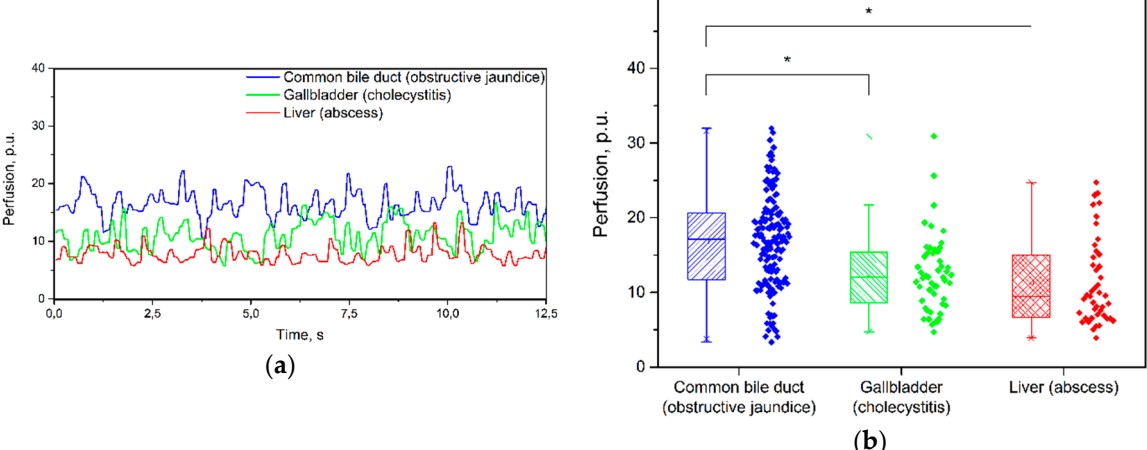

**Figure 4.** Typical LDF signals in the areas of interest (**a**) and Box-Whisker diagram of mean perfusion (**b**). * Confirmed statistically significant differences. The level of significance was estimated using a one-way analysis of variance ($p < 0.05$)

**Table 2.** Average values of LDF signals.

| Area | Mean | Standard Deviation |
|---|---|---|
| Common bile duct | 16.82 | 6.24 |
| Gallbladder | 12.46 | 5.03 |
| Liver | 11.34 | 5.72 |

The average perfusion in common bile duct significantly exceed the values of this parameter in the other areas, as the fluorescence spectra does. Usually, it is expected that fluorescence intensity decreases as blood perfusion increases and more optical radiation is absorbed, especially in the range of 400–600 nm. However, experimental measurements showed that common bile duct has higher fluorescence intensity as well as blood perfusion. This can relate with the influence of the factors mentioned earlier. Nevertheless, both methods used shows the sensibility to the area of interest and its pathology.

So far, laser Doppler flowmetry method has not been widely used in surgery, since there are a number of methodological difficulties for that. The most important one is the difficulty in assessing the effect of probe pressure on recorded signal. In addition, a few researchers of abdominal organs microcirculation use equipment of different manufacturers. Nevertheless, earlier studies confirm the fact that the use of laser Doppler flowmetry in multimodal optical devices for examining the gastroduodenal area is a useful addition in postoperative observation of patients with gastrointestinal diseases, since no other methods provide a continuous evaluation of perfusion [23,24]. An example of the analysis of pathophysiological changes in blood flow is known in studies of benign and malignant neoplasms in the stomach in vivo [25]. The examples show that in the long term an evaluation of blood perfusion of the tissue can become an additional diagnostic tool for assessing tissue viability, which will allow a surgeon to separate tissues by their state of vitality during operations more accurately.

## 4. Conclusions

The conducted studies show the possibility of combined using of FS and LDF methods in minimally invasive surgery of abdominal organs. The obtained preliminary data demonstrated statistically significant differences in fluorescence intensity maxima that proves the sensitivity of proposed approach to the properties of the area of interest as well as presence of pathological changes. Moreover, the differences were observed more clearly for excitation wavelength of 450 nm ($1.43 \pm 0.61$ a.u. for common bile duct and $0.93 \pm 0.49$ a.u. for gallbladder; $0.63 \pm 0.39$ a.u. for liver) rather than for excitation wavelength of 365 nm ($0.90 \pm 0.36$ a.u. for common bile duct and $0.66 \pm 0.34$ a.u. for gallbladder). The analysis of LDF signals showed similar results as well ($16.82 \pm 6.24$ p.u., $12.46 \pm 5.03$ p.u. and $11.34 \pm 5.72$ for common bile duct, gallbladder and liver, respectively). At the same time, it was concluded that further development of this topic require more detailed studies of variety of influencing factors. Therefore, it is planned to further optimize the technical part and develop new diagnostic criteria based on the results of further research. In total, suggested multimodal approach seems promising and can be used as instrumental method for controlling tissues and mucous membranes in abdominal organs.

**Author Contributions:** A.D. funding acquisition, supervision; A.M., A.D. conceptualization; E.P., E.Z., V.D. methodology, formal analysis; K.K., A.M., A.A. investigation; A.M., A.A., Y.I., A.B. resources; E.P., K.K. writing—original draft preparation; E.Z., V.D., A.D. writing—review and editing.

**Funding:** This research was funded by the Russian Science Foundation under grant number 18-15-00201.

**Acknowledgments:** Special thanks are extended to the patients of the Orel Regional Clinical Hospital, which kindly agreed to take part in the studies in the frame of taking their planned minimally invasive surgical intervention.

**Conflicts of Interest:** The authors declare no conflict of interest. The funders had no role in the design of the study; in the collection, analyses, or interpretation of data; in the writing of the manuscript, or in the decision to publish the results.

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
