# Peer review of "Fiber-Optic System for Intraoperative Study of Abdominal Organs during Minimally Invasive Surgical Interventions"

_applsci, doi:10.3390/app9020217_

Round 1
Reviewer 1 Report
Page 2 Line 87-91: Re-write this paragraph, its unclear: “In this study, we hypothesized that FS and LDF combined together to get a highly sensitive multimodal instrument for the study of tissue vitality can be successfully used during minimally invasive interventions [15–17]. Thus, the purpose of this work was to study the possibilities of the technical implementation of FS and LDF in one experimental setup and intra-operrative combined application of these methods in pathology of the organs of hepatopancreatoduodenal zone.”
Page 3 Line 93: Replace the author created acronyms FS and LDF by their full forms to avoid readers going back and forth to find out the meaning of these acronyms.
Page 3 Line 96-98: The visibility of this figure is poor. Plots in the laptop are not readable. Also, increase the fonts and size of Fig 1(a). Fig 1(b) is fine.
Page 4 Line 137-140: Improve the figure quality, such as border lines, legends, labels, and so on. In overall, authors need to find a way to improve the quality of their figures.
Page 5 Lin 179: At the end of results, authors should avoid using acronyms, LDF, instead use their full name.
Page 5 Line 184: Conclusion and Summary are two different things. The summary is a restatement of the entire paper while the conclusion is an ending statement. This reviewer finds over 90% of authors seem to have no clear understanding of the distinction that exists between the terms “Conclusion and Summary”. Please remove the term “To sum up” here.
This manuscript does not include any data table. No concrete numbers are given in the abstract and conclusion section although the result section seems to include some. This makes the paper weak. I encourage authors to support the evidence using numbers or percentage, especially the conclusion section.
Author Response
We would like to thank you for your constructive feedback to improve the quality of our manuscript. We have carefully considered each suggestion and amended the paper accordingly. We include below a list of the comments and our answers, as well as the corresponding changes made to the paper.
Point 1: Page 2 Line 87-91: Re-write this paragraph, its unclear: “In this study, we hypothesized that FS and LDF combined together to get a highly sensitive multimodal instrument for the study of tissue vitality can be successfully used during minimally invasive interventions [15–17]. Thus, the purpose of this work was to study the possibilities of the technical implementation of FS and LDF in one experimental setup and intra-operrative combined application of these methods in pathology of the organs of hepatopancreatoduodenal zone.”
Response 1: We have re-written this paragraph (Page 2, Lines 95-99):
In this study, we have combined FS and LDF measurement techniques in one fibre optical probe to record the parameters of endogenous fluorescence and blood perfusion during minimally invasive interventions [15–17]. Thus, the overall aims of this work were to test the technical implementation of such measurements as well as register reference patterns of the fluorescence spectra and blood perfusion time series in organs of hepatopancreatoduodenal zone.
Point 2: Page 3 Line 93: Replace the author created acronyms FS and LDF by their full forms to avoid readers going back and forth to find out the meaning of these acronyms.
Response 2: We agree that frequent use of acronyms can confuse readers. FS and LDF have been replaced (Page 2, Line 101).
Point 3: Page 3 Line 96-98: The visibility of this figure is poor. Plots in the laptop are not readable. Also, increase the fonts and size of Fig 1(a). Fig 1(b) is fine.
Response 3: We see that labels in Figure 1 (a) were hard to read; therefore, the fonts have been enlarged. The resolution of the figure itself have been increased. We also have increased font size in Figure 1 (b). To make both figures more readable we have placed Figure 1 (b) under Figure 1 (a) (Page 3).
Point 4: Page 4 Line 137-140: Improve the figure quality, such as border lines, legends, labels, and so on. In overall, authors need to find a way to improve the quality of their figures.
Response 4: We have increased the resolution of Figures 2 (a, b), 3 (a, b) and 4 (a, b), as well as made brighter colours and more visible borders (Pages 4-6).
Point 5: Page 5 Lin 179: At the end of results, authors should avoid using acronyms, LDF, instead use their full name.
Response 5: We understand that it was inconvenient to use acronyms at the end of Results and Discussion section. We have removed the acronym LDF in Page 6, Lines 196, 200.
Point 6: Page 5 Line 184: Conclusion and Summary are two different things. The summary is a restatement of the entire paper while the conclusion is an ending statement. This reviewer finds over 90% of authors seem to have no clear understanding of the distinction that exists between the terms “Conclusion and Summary”. Please remove the term “To sum up” here.
Response 6: Indeed, the Conclusion section looked more like Summary rather than Conclusion and it has lacked certain data. “To sum up” have been removed and Conslusion have been extended (additional data included in Page 6, Lines 210-217):
The obtained preliminary data demonstrated statistically significant differences in fluorescence intensity maxima that proves the sensitivity of proposed approach to the properties of the area of interest as well as presence of pathological changes. Moreover, the differences were observed more clearly for excitation wavelength of 450 nm (1.43±0.61 a.u. for common bile duct and 0.93±0.49 a.u. for gallbladder; 0.63±0.39 a.u. for liver) rather than for excitation wavelength of 365 nm (0.90±0.36 a.u. for common bile duct and 0.66±0.34 a.u. for gallbladder). The analysis of LDF signals showed similar results as well (16.82±6.24 p.u., 12.46±5.03 p.u. and 11.34±5.72 for common bile duct, gallbladder and liver, respectively).
Point 7: This manuscript does not include any data table. No concrete numbers are given in the abstract and conclusion section although the result section seems to include some. This makes the paper weak. I encourage authors to support the evidence using numbers or percentage, especially the conclusion section.
Response 7: We agree that the data, which the plots were based on, is necessary to prove the results we obtained. Considering this recomendation, we have included following two tables representing the data in Figures 3 and 4 (b) (Pages 5-6, Lines 178 and 186).
Table 1. Averaged fluorescence intensity maxima.
Area | Mean | Standard Deviation | ||
365 nm | 450 nm | 365 nm | 450 nm | |
Common bile duct | 0.90 | 1.43 | 0.36 | 0.61 |
Gallbladder | 0.66 | 0.93 | 0.34 | 0.49 |
Liver | 0.67 | 0.63 | 0.45 | 0.39 |
Table 2. Average values of LDF signals.
Area | Mean | Standard Deviation |
Common bile duct | 16.82 | 6.24 |
Gallbladder | 12.46 | 5.03 |
Liver | 11.34 | 5.72 |
The numbers have been included in the Conclusion as well (see Response 6).

Reviewer 2 Report
In this manuscript, the authors have developed a fiber-based setup for nondestructive laparoscopic measurements of in vivo diagnostics. The device is consisted of one fluorescence spectroscopy and laser Doppler flowmetry to detect the endogenous fluorescence and blood perfusion. In combination of the use of fiber optical probes, in vivo detection of the pathology of the organs of hepatopancreatoduodenal area of clinical patients was achieved. This work looks interesting and promising. I recommend the paper for publishing in this journal after the below concerns have been addressed:
Comments:
1. In Page 2, Line 87, the authors have mentioned that “In this study, we hypothesized that FS and LDF combined together to get highly sensitive multimodal instrument for the study of tissue vitality can be successfully used during minimally invasive interventions [15–17]. Thus, the purpose of this work was to study the possibilities of the technical implementation of FS and LDF in one experimental setup and intraoperative combined application of these methods in pathology of the organs of hepatopancreatoduodenal zone.” However, they did not mention what are the advantages of their technique reported here in comparison to those current state-of-art non-invasive diagnostic approaches. For example, magnetic imaging and CT imaging.
2. In Page 3, Figure 1, “Schematic presentation of installation for experimental studies (a) and location of fibers in the laparoscopic probe (b)”. The authors have chosen the excitation light sources 365 nm and 450 nm, which are located within or close to the UV region. Light beams at these wavelengths are known to be harmful to the normal cell and tissues. The authors should explain why they selected these two specific wavelengths and their potential risk during the detection process.
3. In Page 4, Line 130, the authors described that” The FS spectra obtained during the measurements were normalized by the level of backscattered excitation radiation. The results of averaging these spectra are shown in Figure 2. The main feature of the collected data is a high variability of fluorescence intensities. It is assumed that this is caused by both individual variability and different factors, such as, the state and nature of changes in tissues, the phase development of pathological processes as well as ongoing treatment. In some patients, it was difficult to register fluorescence spectra with sufficient level of signal to noise ratio, which may be due to the presence of bile, blood and pus, which absorbed major part of fluorescence emission [20,21]. Such effect is also needed to be taken into account for correct interpretation of obtained data” Here, the authors did not discuss how they could eliminated the environmental noise or other interference signals induced for the fluorescent signals. And how to improve the signal to noise ratio for the device itself or any possibility to develop a reference signal channel.
Author Response
We deeply appreciate your valuable questions and comments. We have revised the manuscript according to the comments and provide below our replies to the comments as well as the changes in the manuscript itself.
Point 1: In Page 2, Line 87, the authors have mentioned that “In this study, we hypothesized that FS and LDF combined together to get highly sensitive multimodal instrument for the study of tissue vitality can be successfully used during minimally invasive interventions [15–17]. Thus, the purpose of this work was to study the possibilities of the technical implementation of FS and LDF in one experimental setup and intraoperative combined application of these methods in pathology of the organs of hepatopancreatoduodenal zone.” However, they did not mention what are the advantages of their technique reported here in comparison to those current state-of-art non-invasive diagnostic approaches. For example, magnetic imaging and CT imaging.
Response 1: We agree that the features of other techniques should be described as well. In the manuscript we mostly described conventional biopsy in comparison with optical one as the purpose of optical methods implementation in minimally invasive surgery is closer to biopsy. However, we agree that readers can be confused by the question why other currently used methods are not mentioned in the manuscript. Therefore, we have included recommended comparison with methods of X-ray and ultrasound examination, MRI and CT (additional data included in Page 2, Lines 55-64):
Currently, the integral parts of any minimally invasive intervention are the methods of ultrasound and X-ray examination. They are necessary for determining the allocation of pathological formations (such as neoplasms or abscesses), as well as more complex methods of computer tomography (CT) and magnet resonance imaging (MRI). However, these methods mostly show anatomical and morphological features at the organ level whereas in most cases it is important to assess the type and vitality of tissues at the cellular level. Therefore, obtaining of additional diagnostic information about the properties of the tissues intraoperatively is of particular interest for modern surgery. Speaking of CT and MRI these methods cannot be used immediately during the surgery. Another problem of X-ray and CT methods to be worried about is the harmful effects of the radiation.
Point 2: In Page 3, Figure 1, “Schematic presentation of installation for experimental studies (a) and location of fibers in the laparoscopic probe (b)”. The authors have chosen the excitation light sources 365 nm and 450 nm, which are located within or close to the UV region. Light beams at these wavelengths are known to be harmful to the normal cell and tissues. The authors should explain why they selected these two specific wavelengths and their potential risk during the detection process.
Response 2: We understand the concerns regarding the application of UV radiation. We have provided more detailed explanation on the safety of LED source of near UV radiation we used in the experimental set-up (Page 3, Lines 120-125).
To assess the safety of the optical probe, effective tissue irradiance was calculated and its value was about 1.4 W/m2. According to the guidelines of the International Commission on Nonionizing Radiation Protection (ICNIRP), the limiting UV exposure duration per day for this level of effective irradiance is at least 10 s. In the study, the actual time of the UV-irradiation did not exceed 2 s. [19]. Therefore, the using of proposed source wavelength was deemed safe. The output power for an excitation wavelength of 450 nm does not exceed 3.5 mW.
Point 3: In Page 4, Line 130, the authors described that” The FS spectra obtained during the measurements were normalized by the level of backscattered excitation radiation. The results of averaging these spectra are shown in Figure 2. The main feature of the collected data is a high variability of fluorescence intensities. It is assumed that this is caused by both individual variability and different factors, such as, the state and nature of changes in tissues, the phase development of pathological processes as well as ongoing treatment. In some patients, it was difficult to register fluorescence spectra with sufficient level of signal to noise ratio, which may be due to the presence of bile, blood and pus, which absorbed major part of fluorescence emission [20,21]. Such effect is also needed to be taken into account for correct interpretation of obtained data” Here, the authors did not discuss how they could eliminated the environmental noise or other interference signals induced for the fluorescent signals. And how to improve the signal to noise ratio for the device itself or any possibility to develop a reference signal channel.
Response 3: Indeed, we discussed the influence of various factors on the data obtained but did not mentioned the measures to eliminate or at least reduce their influence. Considering the recommendation and possible interest of future readers in this question, the possible approaches to solve methodological problems have been included in the following paragraph (Page 4, Lines 152-165):
The components of blood, especially haemoglobin, are good absorbers of optical radiation and one of possible measures to reduce this effect is applying to the probe a moderate pressure. This technique allows one to decrease the blood content in the tissue as well as alleviate the attenuation effect of blood on the fluorescence spectrum [21]. However, in surgery in many cases the pressure applied to the end of optical probe can have negative effects on the soft tissues under study. The problem of accurate measurements requires an appropriate solution. The use of additional optical methods (e.g. diffuse reflectance spectroscopy) can be considered. Extending the number of implemented modalities allows for obtaining the data, which can be used to compensate the influence of blood on recorded fluorescence spectra [22]. If the blood content of the living tissue is to be kept unaffected, then to eliminate the probe pressure, an embedded in the probe noncontact distance measurements with subsequent automation of the recording procedure at a fixed optimal distance can be one of the possible solutions. Moreover, another measure to increase the level of signal to noise ratio is automation of mucus, pus and blood removal from the surface of optical probe during the procedure with minimal impact on the surface of biological tissue.

Round 2
Reviewer 2 Report
The manuscript has been improved after revisions according to the reviewers' comments. I recommend the paper to be published after minor revision with careful English grammar checking.